# 3D Printing of a Reactive Hydrogel Bio-Ink Using a Static Mixing Tool

**DOI:** 10.3390/polym12091986

**Published:** 2020-08-31

**Authors:** María Puertas-Bartolomé, Małgorzata K. Włodarczyk-Biegun, Aránzazu del Campo, Blanca Vázquez-Lasa, Julio San Román

**Affiliations:** 1Institute of Polymer Science and Technology, ICTP-CSIC, Juan de la Cierva 3, 28006 Madrid, Spain; mpuertas@ictp.csic.es (M.P.-B.); jsroman@ictp.csic.es (J.S.R.); 2CIBER’s Bioengineering, Biomaterials and Nanomedicine, CIBER-BBN, Health Institute Carlos III, Monforte de Lemos 3-5, 28029 Madrid, Spain; 3INM—Leibniz Institute for New Materials, Campus D2 2, 66123 Saarbrücken, Germany; m.k.wlodarczyk@rug.nl (M.K.W.-B.); aranzazu.delcampo@leibniz-inm.de (A.d.C.); 4Chemistry Department, Saarland University, 66123 Saarbrücken, Germany

**Keywords:** 3D-bioprinting, static mixer, reactive hydrogel, chitosan, hyaluronic acid

## Abstract

Hydrogel-based bio-inks have recently attracted more attention for 3D printing applications in tissue engineering due to their remarkable intrinsic properties, such as a cell supporting environment. However, their usually weak mechanical properties lead to poor printability and low stability of the obtained structures. To obtain good shape fidelity, current approaches based on extrusion printing use high viscosity solutions, which can compromise cell viability. This paper presents a novel bio-printing methodology based on a dual-syringe system with a static mixing tool that allows in situ crosslinking of a two-component hydrogel-based ink in the presence of living cells. The reactive hydrogel system consists of carboxymethyl chitosan (CMCh) and partially oxidized hyaluronic acid (HAox) that undergo fast self-covalent crosslinking via Schiff base formation. This new approach allows us to use low viscosity solutions since in situ gelation provides the appropriate structural integrity to maintain the printed shape. The proposed bio-ink formulation was optimized to match crosslinking kinetics with the printing process and multi-layered 3D bio-printed scaffolds were successfully obtained. Printed scaffolds showed moderate swelling, good biocompatibility with embedded cells, and were mechanically stable after 14 days of the cell culture. We envision that this straightforward, powerful, and generalizable printing approach can be used for a wide range of materials, growth factors, or cell types, to be employed for soft tissue regeneration.

## 1. Introduction

3D bioprinting is a booming additive manufacturing technology that allows the layer-by-layer deposition of a cell-laden material to fabricate 3D constructs with spatial control over scaffold design. This technology has been widely used in the last few years for tissue engineering and regenerative medicine applications as it allows the artificial reconstruction of the complexity of native tissues or organs [1,2,3]. To date, great efforts have been made to develop suitable bio-inks to provide cell-laden scaffolds with good mechanical properties as well as high cell viability. Hydrogels are often used as supporting material in the bio-ink due to their favorable intrinsic properties for supporting cellular growth [4,5,6,7,8,9,10,11,12,13]. Their unique inherent properties similar to the extracellular matrix (EMC), such as porosity that allows nutrient and gaseous exchange, high water content, biodegradability, and biocompatibility make them attractive for cell therapy applications [14,15,16]. Specifically, bio-inks based on natural hydrogels, such as alginate, agarose, gelatin, collagen, chitosan, or hyaluronic acid are regularly used [4,17,18,19,20,21,22,23].

Extrusion is the most frequently used technique for 3D bio-printing with hydrogel precursors [24,25]. The minimal requirements that the hydrogel bio-ink has to fulfil for successful extrusion include: (1) bio-ink must easily flow through the needle during printing but retain the shape after extrusion, (2) printed strands should have a good structural integrity to provide self-supporting structures with good adhesion between layers, and (3) bio-ink must ensure cell survival and proliferation within the printed scaffold [26,27]. Naturally derived hydrogels are still challenging to print due to their weak mechanical properties that lead to poor printing accuracy and low stability of the printed structures [9,13,28,29,30,31]. Traditional approaches based on increasing the polymer content and viscosity or the crosslinking density have been attempted to improve printability of naturally derived hydrogel bio-inks and the mechanical performance of their printed scaffolds [5,27]. However, bio-inks with high polymer contents or viscosities can compromise cell viability due to the high shear forces and lower nutrients transport through the printed constructs [5,27,32]. Thus, the development of low viscosity bio-inks and suitable printing extrusion processes for bio-fabrication are still in demand.

In this paper, we present a methodology for printing homogeneous strands from a reactive hydrogel using a dual syringe system with a static mixing tool coupled to an extrusion bio-printer. The two reactive hydrogel precursors are loaded into separate syringes, simultaneously extruded by mechanical displacement and transported to the static mixer in a 1:1 ratio. They are homogeneously mixed during the short residence time in the static mixer and the crosslinking reaction is initiated prior to extrusion. The partially crosslinked hydrogel is then extruded from the printhead. This approach has several advantages for 3D extrusion printing: (1) it uses low viscosity starting solutions of the hydrogel precursors and avoids high shear stress during extrusion, (2) the in situ crosslinking provides appropriate structural integrity to the printed thread to maintain the printed shape, (3) it avoids post-printing cell seeding (cells are embedded in the ink), washing steps, or additional physical factors that may complicate the fabrication process [28].

Two component systems have been used for 3D printing. Skardal et al. [33] used methacrylated gelatin and methacrylated hyaluronic acid [33] to print scaffolds with gradient material properties, but exchange of the syringes during the printing process was required. Bakarich et al. [24] and O’Connell et al. [34] presented a two component extrusion printing system (alginate/polyacrylamide and an acrylated urethane [24], or gelatin–methacrylamide and hyaluronic acid–methacrylate [34]) where materials were mixed prior to printing in a static nozzle or in the needle, and UV irradiation was required for stabilization of the printed structures. Reactive hydrogels have been used in 3D printing. Gregor et al. [35] and Zimmermann et al. [36] prepared 3D scaffolds by fusing individual droplets of two precursors (fibrinogen and thrombin [35] or thiol-terminated starPEG and maleimide-functionalized heparin [36]), and Lozano et al. [37] used a hand-held system with a coaxial syringe tip to extrude precursors (gellan gum-RGD and CaCl_2_). However, mixing and crosslinking take place during droplet deposition, and scaffolds with spatially graded material properties were obtained [36,37]. Maiullari et al. [38] used a microfluidic 3D printing approach where alginate/PEG-fibrinogen and CaCl_2_ precursors were mixed after extrusion from a coaxial needle and an ultraviolet (UV) crosslinking step was needed to increase stability of the printed structure. Bootsma et al. [39] used a mixing head in order to combine hydrogel precursors (alginate/acrylamide/*N,N*-methylenebisacrylamide/d-glucono-δ-lactone and alginate/CaCO_3_/Irgacure 1173) even though additional UV crosslinking was still necessary to induce covalent crosslinking. On the other hand, the static mixing tool has been used for different reactive hydrogel systems in several reported works. Deepthi et al. prepared an injectable fibrin hydrogel containing alginate nanobeads using a double syringe connected to the static mixer [40]. Hozumi et al. studied the gelation process through a static mixer of alginate hydrogels with Ca^2+^ [41]. In addition, different studies have reported the use of static mixers in other hydrogel processing technologies, like in injectable [42,43,44,45] or moulding [46,47] formulations, but, in all these examples, shape fidelity has not been addressed. To our best knowledge, static mixers have not been explored for 3D bioprinting of reactive hydrogels with good shape fidelity and resolution.

In order to take advantage of a static mixer for 3D printing of a reactive two-components ink, the gelation kinetics has to be carefully adjusted to the residence time in the tool. In this work, we used the naturally derived polysaccharides chitosan [28,48,49,50] and hyaluronic acid [21,51,52,53] modified with reactive groups and formulated in two separate precursor solutions. Chitosan is a great candidate for tissue engineering applications since it exhibits notable biological features such as great cytocompatibility and biodegradability, antibacterial, hemostatic, or muco-adhesive properties [54,55]. Specifically, a carboxymethyl chitosan derivative (CMCh) was selected because of its good solubility at physiological pH, which allows straightforward encapsulation of cells in the bio-ink [56,57] and avoids any neutralization or washing steps, commonly used for pure chitosan-based printing [28,49,58,59]. Hyaluronic acid (HA) is an important component of the extracellular matrix (ECM), which favors cell affinity and proliferation [60,61]. HA is not suitable by itself for 3D printing, but it can improve printability of the bio-ink due to its shear-thinning behavior [52], and, when it is in combination with chitosan, it can counteract the brittle mechanical properties of the former [54]. Thus, when using a CMCh/HA ink, the free amines of CMCh can react with the aldehyde groups of partially oxidized HAox [62] after mixing via Schiff base formation [56,63,64,65], which gives rise to a hydrogel structure. This system has been demonstrated to allow viable cell encapsulation [66]. In this paper, this printing methodology has been optimized in order to fulfil the requirements for successful bioprinting to lead to cell laden 3D hydrogel constructs with good resolution and shape fidelity.

## 2. Materials and Methods

### 2.1. Materials

Carboxymethyl chitosan (CMCh) (degree of deacetylation 85–90%, viscosity = 5–300 mPas, Chitoscience, Halle (Saale, Germany), sodium hyaluronan (HA) (~1.5–1.8 × 10^6^ Da, Sigma-Aldrich, St. Louis, MO, USA), sodium periodate (NaIO_4_) (Alfa Aesar, Haverhill, MA, USA), ethylene glycol (Sigma), hydroxylamine (Sigma-Aldrich), iron chloride (III) (Sigma-Aldrich), and phosphate buffered saline solution (PBS) 10 mM pH 7.4 (Gibco, Thermo Fisher, Waltham, MA, USA) were used as received. Sodium hyaluronan of low molecular weight (M_w_ ~200 kDa, Bioiberica, Barcelona, Spain) was oxidized (HAox) prior to use, as reported elsewhere [62], with a final oxidation degree of 48 ± 3.2% [67,68].

### 2.2. Bio-Ink Formulation

Hydrogel inks were formulated in two separate solutions. CMCh was dissolved in PBS (pH = 7.4) unless otherwise noted. HAox or HAox with HA mixture was dissolved in 0.1 M NaCl. The final ink formulation is named CMChn/HAoxn-HAn, where the number (n) that follows the polysaccharide abbreviation means the weight percentage of the precursor solution. Initially, different CMCh/HAox and CMCh/HAox-HA compositions were tested for optimization of the printing process. Lastly, 3D printed scaffolds were prepared with the optimized hydrogel formulation: CMCh2/HAox4-HA0.4. A 1:1 volume ratio of the two solutions was used for printing. To increase in vitro stability of the printed scaffolds for longer term cell studies, a post-printing stabilization step was carried by immersion in a 20 mM FeCl_3_ aqueous solution for 7 min.

### 2.3. 3D Printing with a Static Mixing Tool

The 3D Discovery printer (RegenHu, Villaz-Saint-Pierre, Switzerland) was modified to accommodate a static mixing tool. The mixing tool system consists of two 1-mL disposable syringes coupled to a single disposable static mixer (2.5 mL length, helical screw inside) provided by RegenHu Company, and a printing needle (Figure 1A). To employ the mixing system, the original high precision plunger dispenser of the printer was adjusted with a custom-made holder for the static mixing tool. The obtained dual extrusion printing head employs simultaneous mechanically-driven movement of two syringe plungers using a single motor (Figure 1B). This leads to a 1:1 mixing ratio of the liquids from the connected syringes that have the same dimensions. The plungers’ speed is controlled by the software, and accordingly modified by RegenHu. The solutions are transported to the static mixer, and then extruded through the connected needle (Figure 1C).

Printing and plunger speeds were optimized for each tested ink formulation. Values in the range of 5 to 25 mm/s (print head movement speed) and 0.04 to 0.1mm/s (plunger speed) were tested, using a design of 4-cm long parallel lines. A conical polyethylene needle with an inner diameter of 200 μm was used. 3D scaffolds (2 or 4 layer grid square: 12 × 12 mm^2^, 1.5 mm separation between strands) were printed with 15 mm/s printing speed and 0.06 mm/s plunger speed. Prior to start of the designed architecture printing, one sacrificial 4-cm long line was printed to allow material homogenization in the mixer. Scaffolds of formulations without encapsulated cells were printed onto granulated paper (hp laserjet transparency film) and cell-laden scaffolds of the optimized formulation CMCh2/HAox4-HA0.4 were printed onto glass coverslips.

### 2.4. Bio-Ink and 3D Printed Scaffolds Characterization

#### 2.4.1. Rheological Analysis

Rheology experiments were performed at a controlled temperature of 25 °C, using a rheometer (ARG2, TA Instruments, New Castle, DE, USA) equipped with a parallel plate sand-blasted geometry (25-mm diameter).

Gelation times were measured in oscillation mode in time sweep experiments. Storage modulus (G′) and loss modulus (G″) were recorded at a frequency of 1 Hz and 1% strain over 5 min. Furthermore, 75 μL of CMCh solution was deposited on the lower plate. Then, the same volume of the corresponding HAox or HAox-HA solution was deposited on top, shortly after being mixed by pipetting, and the mixture was immediately compressed between measuring plates (300 µm measuring gap). Different compositions were tested: CMCh1/HAox1, CMCh1/HAox2, CMCh2/HAox2, CMCh2/HAox4, CMCh3/HAox3, CMCh3/HAox6, and CMCh2/HAox4-HA0.4. Gelation time was defined as the time when G’ crossed-over G″. Each sample was measured three times and the average gelation time value was given. Additionally, to determine final viscoelastic properties, a frequency sweep experiment was performed for the optimized bio-ink formulation CMCh2/HAox4-HA0.4. Storage modulus (G′) and loss modulus (G″) were recorded at 1% strain and increasing frequencies from 1 to 200 Hz.

Viscosities of 2 wt % CMCh, 4 wt % HAox, and HAox-HA blends (4 wt % HAox with 0.2, 0.4, or 0.6 wt % HA amounts) solutions were determined by rotational shear measurements at an increasing shear rate from 1 to 500 s^-1^. Final viscosity of CMCh2/HAox4-HA0.4 hydrogel was measured at shear rates increasing from 0.1 to 150 s^-1^.

Each sample was measured three times and the average and standard deviation were given.

#### 2.4.2. Attenuated Total Reflection Fourier Transform Infra-Red (ATR-FTIR) Spectroscopy

Attenuated Total Reflection Fourier Transform Infra-Red spectra of CMCh, HAox-HA, and CMCh2/HAox4-HA0.4 samples were measured for structural characterization (ATR-FTIR, Perkin-Elmer Spectrum One, Madrid, Spain).

#### 2.4.3. Scaffolds Morphology

Light microscopy characterization of printed formulations was performed with a stereomicroscope SMZ800N (Nikon, Tokyo, Japan) equipped with home-made bottom illumination and camera (13MPx, Samsung, Seoul, Korea) for imaging.

#### 2.4.4. In Vitro Swelling and Degradation Studies

Swelling and degradation in vitro were carried out in physiological (PBS pH = 7.4 at 37 °C) conditions to evaluate stability of just-printed scaffolds, and scaffolds with additional post-printing stabilization. The additional post-printing stabilization step was performed by immersion in a 20 mM FeCl_3_ aqueous solution for 7 min. For the in vitro swelling experiments, printed samples (two layers of square-based scaffolds) were incubated in PBS for 4 h and, after gentle removal of excess of PBS, imaged using a stereomicroscope SMZ800N (Nikon, Dusseldorf, Germany). Swelling was evaluated by measuring strand widths in the scaffold with imageJ software before and after incubation. A minimum of four replicates was analyzed and results were given as mean ±SD. In vitro degradation was qualitatively analyzed by microscope pictures after 1, 4, 7, 14, and 21 days of incubation in PBS. Images were taken using a microscope Nikon Eclipse TE2000-S with camera NikonDS-Ri2.

### 2.5. CMCh2/HAox4-HA0.4 Based Bioprinting

#### 2.5.1. Cell Culture

L929 Fibroblasts (ATCC, Manassas, VA, USA) were cultured in RPMI 1640 phenol red free medium (Gibco, 61870-010) supplemented with 20% fetal bovine serum (FBS, Gibco, 10270), 200 mM L-glutamine, and 1% penicillin/streptomycin (Invitrogen, Thermo Fisher). Incubation was carried out at 37 °C, 95% humidity, and 5% CO_2_. Cell culture media was changed every two days.

#### 2.5.2. Bio-Ink Preparation

L929 Fibroblasts (ATCC) were loaded in the 2 wt % CMCh solution (prepared in RPMI 1640) at a 2 × 10^6^ cells/mL concentration. Solutions of both hydrogel precursors (2 wt % CMCh loaded with L929, and 4 wt % HAox-0.4 wt % HA) were transferred into the printing syringes and two layers of square-based (120 × 120 mm^2^) scaffolds were printed at r.t onto glass coverslips. The final cell concentration in the scaffold was 1 × 10^6^ cells/mL. An additional post-printing stabilization step was carried by immersion in a 20 mM FeCl_3_ aqueous solution for 7 min. After that time, solution was removed and samples were incubated in cell culture media.

#### 2.5.3. Cell Assays and Staining

In order to evaluate cell viability within the hydrogel scaffold over a 14-day period, staining with fluorescein diacetate (FDA) and propidium iodide (PI) was carried out to detect live and dead cells, respectively. In brief, scaffolds were washed with PBS at different time points (1, 4, 7 and 14 days) and incubated with 20 μg/mL FDA and 6 μg/mL PI for 10 min at r.t. Then, samples were washed with PBS 3× and fluorescence images were taken using Nikon Ti-Ecllipse microscope (Nikon Instruments Europe B.V., Amsterdam, The Netherlands). Quantification was performed by image analysis using the Image-J 1.52p software counting both green and red cells with the function “find maxima.” The cell viability percentage was calculated by quantifying the live cells between the total amount of cells in at least five images for three independent samples. As a control, cell viability studies of cells encapsulated in bulk hydrogels without using the static mixer were performed on the first day. Analysis of variance (ANOVA) using the Tukey grouping method of the results for the printed samples was performed at each time with respect to the first day at a significance level of *** *p* < 0.05 with respect to non-printed samples at a significance level of ^###^
*p* < 0.05.

Fluorescence staining of nuclei was carried out to quantify cell proliferation within the 3D constructs over a 14-day period. Cells were fixed with PFA (paraformaldehyde) 3.7% *w*/*v* for 15 min at different time points (1, 4, 7, and 14 days), which is followed by permeabilization with 0.5% Triton -X 100 in PBS for 15 min and incubated with 1:1000 DAPI (4’,6-diamidino-2-fenilindol) dillution in PBS for 20 min. Lastly, samples were imaged using a LSM 880 confocal microscope (Zeiss, Jena, Germany). Image analysis was performed using the Image-J 1.52p software using the functions “Z project” and “find maxima” to count the number of nuclei observed in all the z levels analyzed by confocal. Cell quantification with ImageJ was performed in three images per sample in a 425.1 × 425.1 µm tested area, and quantification obtained on day 1 was normalized to 100%. Analysis of variance (ANOVA) using the Tukey grouping method of the results for the printed samples was performed at each time point at a significance level of * *p* < 0.05.

## 3. Results and Discussion

### 3.1. 3D Printing with a Static Mixing Tool

The printing conditions to obtain stable threads from the two-component hydrogel system using the static mixer tool were evaluated. Once the two components enter the static mixer, diffusion and a covalent reaction between the amine groups of CMCh and aldehyde groups of HAox is started. Non-covalent interactions such as hydrogen bonds or ionic interactions might further stabilize the gel [69,70,71] and can be beneficial for the extrusion process. For high fidelity printing without clogging the nozzle, the crosslinking kinetics must be adjusted to ensure adequate mixing and good printing quality. The polymer concentration and components ratio, the gelation kinetics, and the viscosity of the inks are relevant parameters to adjust.

The matching between hydrogel crosslinking kinetics and extrusion speed is essential to obtain an adequate crosslinking degree in the static mixer to allow flow while providing smooth and stable strands [39,41]. The crosslinking kinetics for different CMCh and HAox weight concentrations (CMCh1/HAox1, CMCh1/HAox2, CMCh2/HAox2, CMCh2/HAox4, CMCh3/HAox3, and CMCh3/HAox6) were studied in rheological experiments. Figure 2A presents the variation of the shear modulus G′ and the loss modulus G″ as a function of time for the CMCh/HAox formulations that gave a measurable gelation point. As the system began to crosslink through the formation of Schiff base linkages, G′ increased at a faster speed than G″, which indicates a change in the viscoelastic behavior of the system to a more solid-like state. These differential growth speeds led to crossover point of G′ and G″, defined as a gelation point, which indicates that the 3D hydrogel network was formed [66,72]. The corresponding gelation time ranged from 0.90 ± 0.06 to 4.68 ± 0.10 min for the formulations studied (Figure 2B). Regarding ink composition, gelation time decreased with a dropping CMCh/HAox ratio and with an increasing CMCh concentration. The printability of the ink formulations was evaluated by image analysis of printed threads (Figure 2C). Printed threads with 1 wt % CMCh were liquid, which are in agreement with the rheology data that did not show gel formation (undergelation). Broken lines with small gel blocks were visible for 3 wt % CMCh formulations. In these cases, gel formation was faster than the residence time of the solution in the static mixer (over-gelation), and the shear force needed to extrude the ink caused gel fracture. For the intermediate CMCh compositions (2 wt %), semi-solid printed strands were observed, so 2 wt % CMCh was considered a minimum concentration threshold for gel formation. The CMCh2/HAox2 formulation yielded broad lines with low shape fidelity. A feasible region was found for CMCh2/HAox4 formulation, which rendered smooth lines with shape fidelity. In this formulation, the crosslinking degree achieved in the mixing head provided adequate viscosity for extrusion with enough mechanical stability for high fidelity printing. CMCh2/HAox4, with a gelation time of 3.64 ± 0.43 min, was selected as the most appropriate ink for the subsequent experiments.

While printing the CMCh2/HAox4 mixture, bubbles were observed in the needle (Figure 3A, black arrows). In addition, the printed lines had an irregular shape (Figure 3A). We hypothesized that the different viscosities of the precursor solutions due to the different molecular weights of the polymers [64,73] would be the reason for these features. Figure 3B shows that the viscosity of 2 wt % CMCh is 2 orders of magnitude higher than the viscosity of 4 wt % HAox. A different viscosity of the precursor’s solutions is reported to lead to non-homogeneous mixtures due to their different flow through the mixer during extrusion [39,41]. Different strategies have been used in order to adjust viscosities of precursor solutions when using static mixers. For example, Hozumi et al. [41] used carboxymethyl cellulose as a thickening agent, and Bootsma et al. [39] distributed the solution with the largest impact on viscosity in the two syringes. In order to increase the viscosity of the HAox solution, we supplemented it with non-oxidized HA. The viscosity of different HAox-HA blends is also plotted in Figure 3B. All tested solutions presented a shear thinning behavior that facilitates extrusion and shape fidelity [5]. The addition of increasing amounts of HA to the HAox solution lead to a higher viscosity of the mixture. Based on the obtained results, the addition of 0.4 wt % of HA to the 4 wt % HAox solution resulted in a similar viscosity to the 2 wt % CMCh solution. This addition did not influence crosslinking kinetics of the formulation (Appendix A). The printing test with CMCh2/HAox4-HA0.4 formulation (Figure 3C) showed regular and smooth lines without broken parts and no bubbles were formed during the printing process. The CMCh2/HAox4-HA0.4 formulation provided stable filaments with low deviance from the needle geometry and minimized collapsing between the superposed layers visible in the cross-points (Figure 3C). These observations indicate that static mixing of solutions with comparable viscosities improves mixing performance, printing quality, and the resolution.

The printing protocol described in this case allows high fidelity printing of hydrogel structures with low-viscosity ink solutions, which is favorable for cell laden scaffolds [5,27,32]. The hydrogel viscosity, flow rate, and gelation kinetics of the components as they pass through the static mixer affect the mixing performance, homogeneity, and self-support capacity of the bio-ink.

### 3.2. Characterization of the Optimized Bio-Ink

In order to confirm the formation of covalent crosslinks between the CMCh and HAox components of the printing mixture, the CMCh2/HAox4-HA0.4 formulation was characterized by FTIR spectroscopy (see Appendix A). The characteristic peaks corresponding to the functional groups of the CMCh and HAox/HA precursors were observed in the mixture [56,57,62], together with a band at 1653 cm^−1^, which can be attributed to the stretching vibration of the C=N bond of the Schiff base formed by a reaction of amine and aldehyde groups. This indicates that covalent crosslinking was successfully achieved [63,64,74]. Furthermore, a peak was observed at 885 cm^−1^, corresponding to the hemiacetal structure obtained due to the unreacted aldehyde groups of HAox after crosslinking [62]. Intensity of this peak is lower than in the HAox spectrum, which indicates that the rest of the aldehyde groups had participated in the crosslinking reaction.

The viscoelastic properties of the crosslinked CMCh2/HAox4-HA0.4 hydrogel were studied by rheology in frequency sweep experiments. Hydrogel formation was corroborated since storage modulus was always higher than loss modulus. Additionally, a slight frequency-dependent viscoelastic behavior was observed. Presumably, the shear modulus values were mainly due to the covalent crosslinking of the CMCh and HAox functional groups, and HA did not influence crosslinking kinetics or final modulus (Appendix A). Gels were soft with a shear modulus in the range of 50–100 Pa (Figure 4A). This value indicates that these hydrogel scaffolds are promising candidates for regeneration of soft tissues [39], and is comparable to reported chitosan/hyaluronic acid injectable hydrogels with encapsulated cells for abdominal reparation and adhesion prevention [39,63,74,75]. The viscosity of the crosslinked CMCh2/HAox4-HA0.4 bio-ink vs. shear rate is plotted in Figure 4B. The ink viscosity found was relatively low, especially when compared to air pressure-based extruded inks (in the range of 30–6 × 10^7^ mPa) [1,2], which is a desirable feature since low-viscosity bio-inks usually allow higher cell viability [5,27,32]. Solution behaved as a non-Newtonian fluid, where viscosity decreased linearly with an increasing shear rate. This shear thinning behavior is a favorable property for printing. It implies a decrease in the viscosity when the shear stress increases inside the needle under applied pressure, which is followed by a sharp increase of viscosity after extrusion. This facilitates both extrusion and shape fidelity [5].

### 3.3. Characterization of the 3D Printed Hydrogel Scaffolds

2- and 4-layered grid square scaffolds (12 × 12 mm^2^ printed area) were printed using the CMCh2/HAox4-HA0.4 formulation (Figure 5). Good printing accuracy and resolution was observed and stable scaffolds with filaments of uniform dimensions (diameter 357 ± 58 μm) were obtained.

Swelling and degradation rates are relevant parameters when using hydrogels’ bio-inks since they affect the fidelity and stability of the bio-printed scaffolds, as well as allow cellular ingrowth and tissue regeneration [27,76,77]. In this study, swelling and degradability of the printed scaffolds were analyzed by imaging the scaffolds after incubation in PBS for given times and by quantification of the width of the strands (Figure 6A–C). Figure 6A shows the microscopy images of 2-layer printed scaffolds after 4 h of immersion in PBS. A 49 ± 9% swelling was observed under these conditions (Figure 6A). A progressive and notable decrease in scaffold volume was observed with incubation time, up to nearly complete degradation after 7 days (Figure 6B). This degradation rate is slightly faster than previously reported for CMCh/HAox injectable hydrogels (10–14 days) [63,74,75], which can be assigned to the higher surface area and open structure of the grid scaffold that makes them more sensitive for degradation. Schiff’s base crosslinked hydrogels have low stability due to the dynamic nature of the bond [57,78,79]. Thus, to increase the long-term stability of the scaffolds, a new crosslinking approach was proposed since stability is directly related to the crosslinking degree of the hydrogel [76,80,81]. A post-printing crosslinking step was adopted by immersing the scaffold in 20 mM FeCl_3_ for 7 min. Fe (III) forms coordination complexes with hyaluronic acid units at a physiological pH [82], which are expected to act as additional crosslinking points in the printed scaffold. Figure 6A shows lower swelling (19 ± 8%) of the printed scaffold after the second crosslinking step and 4 h after swelling. Additionally, slower degradation of the scaffold was observed after post-printing stabilization (Figure 6C). The scaffold maintained its structural integrity up to 28 days of incubation, although signs of erosion were appreciated in the last stage. In conclusion, the treatment with iron (III) leads to 3D scaffolds with higher structural integrity and long-term stability. This reinforces the stability of Schiff’s base crosslinked hydrogels, which has remained a challenging issue [45,78].

### 3.4. CMCh2/HAox4-HA0.4 Based Bio-Printing

In general, 3D printed hydrogels have been demonstrated to protect cells from mechanical damage induced during the extrusion process, while providing an appropriate environment for the encapsulated cells after printing (by mimicking the ECM) [27,30]. Nevertheless, it is a critical aspect in bio-fabrication to evaluate whether viscosity, in situ crosslinking, and printing process are compatible with encapsulated living cells [4,10]. Thus, the ability of CMCh2/HAox4-HA0.4 formulation to be used as a bio-ink was tested by printing scaffolds with encapsulated L929 fibroblasts. Cell viability in the bio-printed scaffolds was studied during a 14-day period. Live/dead staining allowed imaging of the cells in 2-layer printed scaffolds (Figure 7A). Abundance of cells homogeneously distributed in the scaffold were observed, which reflects the good mixing performance during the printing process. Additionally, cells were released from the hydrogel after 7 and 14 days of culture, which can be due to the degradation rate of the scaffolds. This is a desirable characteristic for potential regenerative approaches for wound healing, where delivered cells would migrate out of the scaffold to heal the injured site [4]. Quantification of cell viability is displayed in Figure 7B, together with data from 3D cultures in non-printed hydrogels of the same formulation as the control. Cells in the printed or non-printed materials showed viability around 60–65% after 1 day of incubation. There are no significant differences between printed and non-printed formulation, which indicates that the printing process did not affect the cells short-term viability. Cell viability in the printed scaffold increased at longer culture times and reached 96% and 95% 7 and 14 days of culture, respectively, which are both significantly different from printed and non-printed formulations after 1 day of incubation. These data suggest that neither the covalent reaction responsible of gelation nor the shear stress produced by the printing process or the stabilization process with iron (III) cause adverse long-term effects on the cells. The proliferation rate of the cells in the scaffolds was analyzed after DAPI staining. Cell proliferation increased over the 14-day period in the printed scaffolds (Figure 7C), and values reached after 14 days of incubation were significantly different from those found after 1 and 4 days of culturing. Lastly, the scaffolds maintained their structural integrity during the whole culture processes (Figure 7D), which indicates that the optimized printed formula and the subsequent stabilization step with iron produced mechanically robust scaffolds with good biocompatibility.

These observations were consistent with other studies based on naturally derived hydrogels bioprinting [17,51,83]. For example, Akkineni et al. studied the encapsulation of endothelial cells in a low viscosity hydrogel core (1% gelatin and 3% alginate) by obtaining cell viability values around 65% one day after printing, which is comparable to our results. A high viscosity shell composed of 10% alginate and 1% gelatin and a secondary crosslinking with CaCl_2_ provided the structural integrity to the scaffold [17]. In addition, Gu et al. presented the direct-write printing of stem cells within a polysaccharide-based bio-ink comprising alginate, carboxymethylchitosan, and agarose. The time course of dead cells content within the optimized bio-ink (containing 5% *w*/*v* of carboxymethylchitosan) demonstrated a relatively high (around 25%) cell death after printing. Subsequently, this decreased to around 10% by day 7, following a trend very similar to that of our work [83]. On the other hand, some reactive hydrogels have been reported for cell encapsulation such as: injectable hydrogels with proliferation trends similar to that found in our work [42], layered platforms with constant cell viability values around 70% until 5 days [37], or gradient formulations [36] where cell viability values slightly decrease with time until around 80% after 7 days. Based on the in-vitro studies, we conclude that the proposed printing technology and bio-ink formulation of this work are suitable as a 3D printing platform for potential biomedical applications as cell carriers in the tissue engineering field.

## 4. Conclusions

The present study describes the development of a reactive hydrogel bio-ink with an extrusion printing methodology based on a dual-syringe system with a static mixing tool. This method shows multiple advantages for 3D extrusion bio-printing. (1) Gelation during the extrusion process provides enough viscosity for printing with good shape fidelity while using low viscosity precursor solutions, (2) the crosslinking during extrusion provides enough structural integrity to retain the printed shape, and (3) the stability of the scaffold, if required for long-term culturing, can be increased in a simple incubation step. Bio-printed scaffolds obtained with our approach showed good biocompatibility, moderate swelling, and shape stability during 14 days of culturing. Since precursors’ concentrations and printing conditions can be easily varied, this printing approach offers high versatility and we envision that it can be adaptable to a wide range of reactive systems with appropriate crosslinking kinetics to be employed in the future for broad applications in regenerative medicine and tissue-engineering.

## Figures and Tables

**Figure 1 polymers-12-01986-f001:**
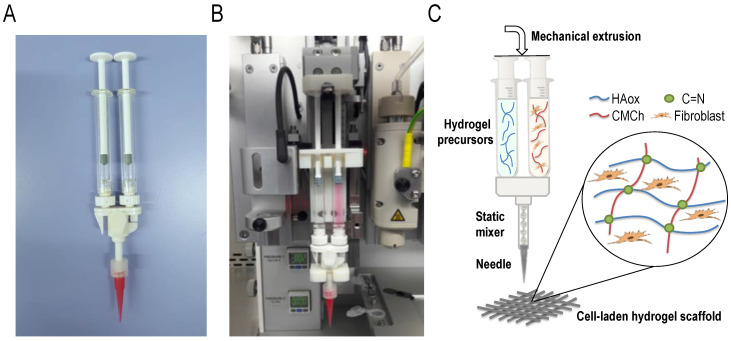
(**A**) Picture of the static mixing tool. (**B**) Static mixing tool coupled to the 3D printer. (**C**) Scheme of the bioprinting process using the mixing device coupled to the 3D printer.

**Figure 2 polymers-12-01986-f002:**
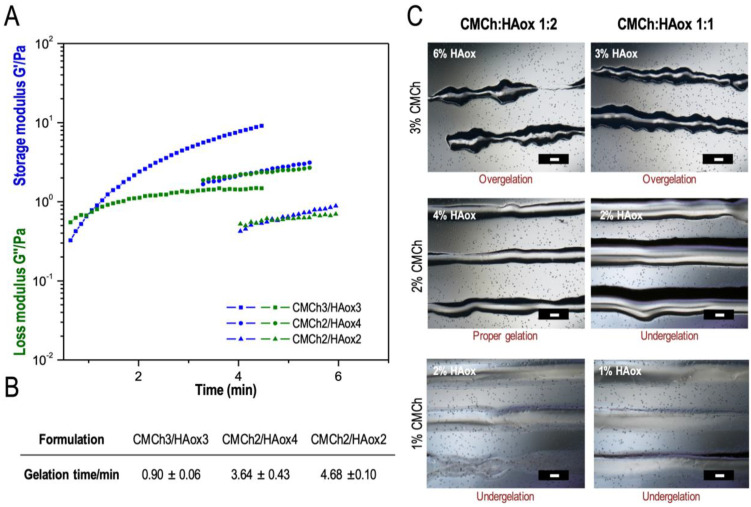
(**A**) Storage (G′) and loss (G″) moduli obtained in time sweep rheological experiments, (**B**) gelation times, defined as G′ and G″ crossover points, and (**C**) light microscope pictures of printed samples of CMCh/HAox formulations with different weight concentrations of CMCh and HAox solutions. Scale bars in a white color correspond to 500 µm.

**Figure 3 polymers-12-01986-f003:**
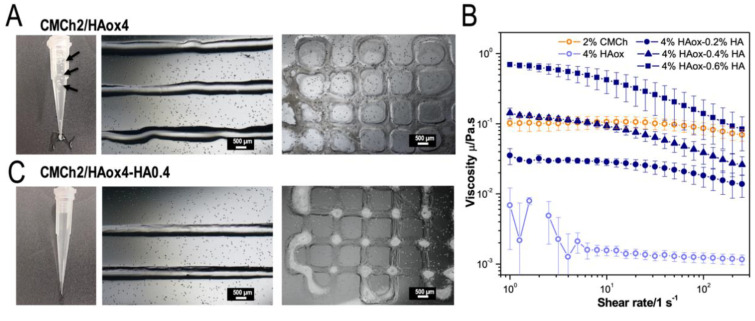
(**A**) Image of the needle during printing. Black arrows highlight bubbles inside the bio-ink. Light microscopy pictures of printed strands and 3D printed scaffolds using CMCh2/HAox4 formulation. (**B**) Viscosity measurements of 2 wt % CMCh, 4 wt % HAox, and different HAox/HA blends. (**C**) Image of the needle during printing and light microscopy pictures of printed strands and 3D printed scaffolds using CMCh2/HAox4-HA0.4 formulation.

**Figure 4 polymers-12-01986-f004:**
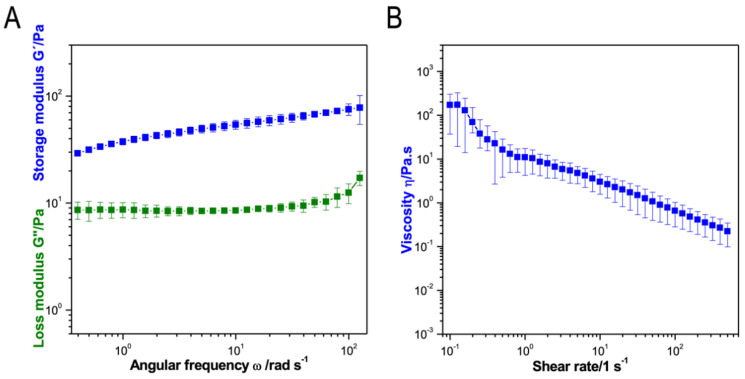
(**A**) Frequency sweep experiment and (**B**) viscosity analysis of CMCh2/HAox4-HA0.4 bio-ink formulation.

**Figure 5 polymers-12-01986-f005:**
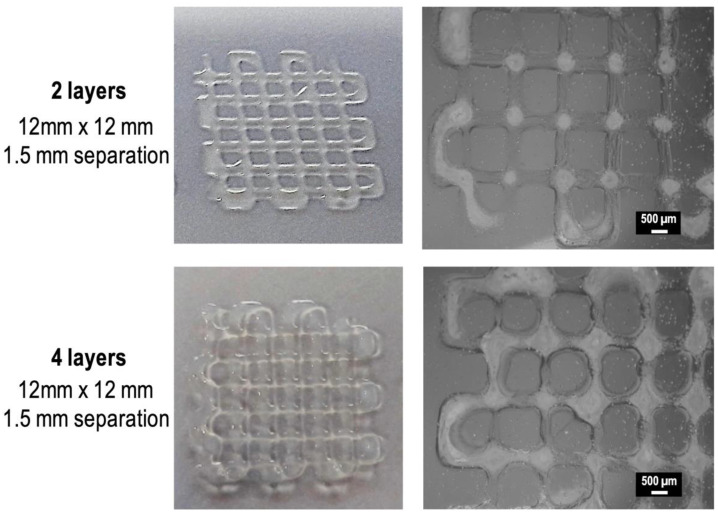
Camera pictures (**left**) and light microscopy pictures (**right**) of 2-layer and 4-layer square scaffolds printed using CMCh2/HAox4-HA0.4 formulation.

**Figure 6 polymers-12-01986-f006:**
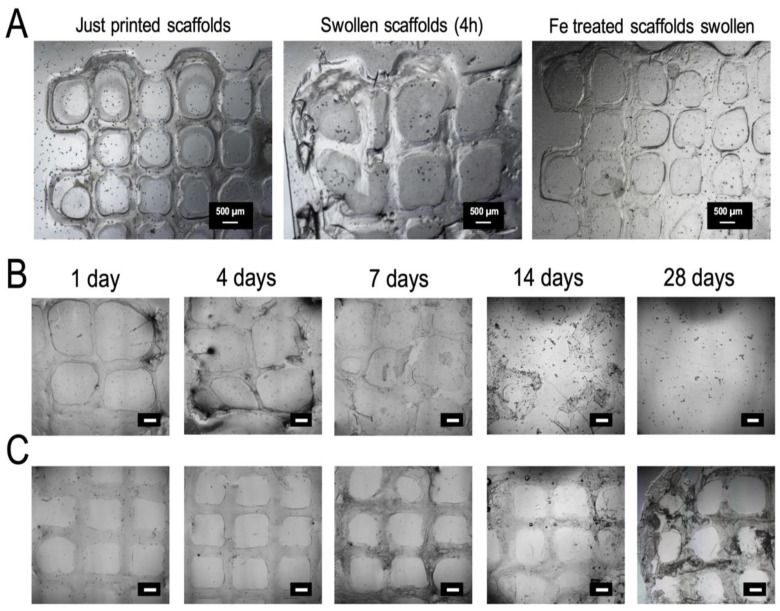
(**A**) Microscopy images of CMCh2/HAox4-HA0.4 printed scaffolds just after printing, after swelling in PBS for 4 h, and after iron treatment and swelling in PBS for 4 h. (**B**) Degradation study of CMCh2/HAox4-HA0.4 scaffolds with no additional treatment after incubation in PBS and (**C**) after iron treatment and incubation in PBS at different time points. Scale bars correspond to 500 µm.

**Figure 7 polymers-12-01986-f007:**
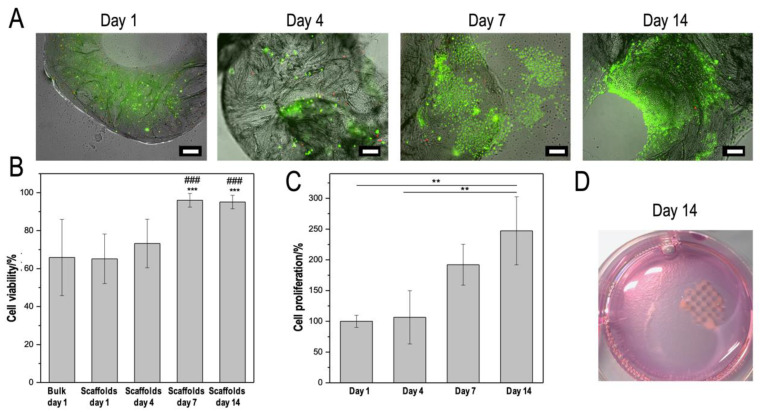
Biological results of CMCh2/HAox4-HA0.4 printed scaffolds loaded with L929 fibroblasts and treated with Fe over a 14-day period: (**A**) Fluorescence imaging of live/dead stained scaffolds at different culture days. (**B**) Quantification of live/dead results including bulk hydrogels at 1 day as a control. Analysis of variance (ANOVA) of the results for the printed samples was performed at each time point with respect to day 1 at a significance level of *** *p* < 0.05, and with respect to non-printed samples at a significance level of ^###^
*p* < 0.05. (**C**) Proliferation assay by quantification of nuclei after DAPI staining. Analysis of variance (ANOVA) of the results for the printed samples was performed at each time point at a significance level of ** *p* < 0.01. (**D**) Picture of a stable printed scaffold after 14 days of incubation. Scale bars correspond to 200 µm.

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
