# Peer review of "3D Printing of a Reactive Hydrogel Bio-Ink Using a Static Mixing Tool"

_polymers, 2020, doi:10.3390/polym12091986_

Round 1

Reviewer 1 Report

This topic is interesting and is suitable to be published in Polymers. There are some comments for revision:

1. The manuscript should be carefully revised, some typos could be observed in the introduction, methods and discussion.
2. The printed hydrogel weight loss (degradation rate) should be investigated in Fig.6 quantitatively. 
3. I think 12mm is right in Fig.5.

4. Why did you choose iron as curing agent instead of other metal ion.

5. Two important references are suggested to cite:

[1] 3D Freeform Printing of Nanocomposite Hydrogels Through in situ Precipitation in Reactive Viscous Fluid, International Journal of Bioprinting 6(2), 2020: 29-49.

[2] 3D printing of hydrogel composite systems: Recent advances in technology for tissue engineering. International Journal of Bioprinting 4 (1), 1-28

Reviewer 2 Report

The authors describe extrusion-based 3D printing of hydrogel bioink that can be mixed via static mixing nozzle and gelled during the printing process. The use of a static mixing tool to enable extrusion-based printing of low-viscosity cell-laden biopolymer scaffold with high cell viability and printing quality is an interesting approach. Overall, the work shows interesting results and analyses that would be the interest of 3D bioprinting community. There are several minor comments on technical details as summarized below:

  1. Mixing of 2 low-viscosity ink during printing through static mixer would be the major point in this work. It would be helpful to have more quantitative characterization of homogeneous mixing either by using fluorescently dyed ink (only one solution, and then quantify homogeneity of fluorescent intensity in mixed & printed inks), etc. The current discussion based on cell dispersion is rather qualitative.

  1. In Figure 4, the storage and shear moduli show that the mixed ink is clearly a gel (higher storage modulus than loss modulus). But viscosity vs shear rate plot shows it shows shear thinning behavior, indicating it is still flowable material. While the proposed method is well described, as indicated in Figure 4, the ink can still be printable after it is fully gelled. It would be very helpful to provide a bit detailed discussion at what point during the printing the ink becomes a gel. Ideally, the ink mixture keeps low viscosity throughout the printing process while passing the gelation point near the tip of the nozzle to minimize shear stress experienced by cells. Did the authors consider or optimize this in the current study?

  1. While the optimal version of the ink shows decent printability, the demonstrated printing resolution is low. It will be helpful if the proposed method also can offer higher resolution printing (~ 100µm diameter etc) of cells while keeping high cell viability.

  1. In Figure 7, why the cell viability of bulk non-printed gel is still relatively low (~ 60 %)? Does this mean the ink formulation itself lowers the cell viability regardless of the printing process?

  1. As a minor point, there are lots of cell-like stuffs (white or black dots) in all printed hydrogel images. What are those dots? Are they really cells? If they are cells, why they disperse in the media rather than well contained within the printed gel scaffold? If 3D printed gel scaffold cannot contain cells in a spatially defined manner, it can be one big disadvantage of the proposed material and printing method.

Reviewer 3 Report

This article on '3D printing of a reactive hydrogel bioink using a static mixing tool' is very well written and is of interest to the additive manufacturing and tissue engineering community. This paper is recommended for publication upon addressing some minor comments as indicated below.

  1. In Sec. 2.4.1, Rheological analysis, please comment on how closely the mixing protocol followed for rheological testing (described in lines 159-160) resembles that of using a static mixer in the printer. Does this replicate the same level of mixing as the 3D printing process? If not, please explain how the gel time measured here is relevant to the actual gel time observed during processing. 
  2. For the rheological test data presented in Fig 2A, Fig 3B and Fig 4 (A and B), please indicate if the charts represent average values from multiple tests. Also, provide error bars on the charts if possible or the percentage error values for these charts in the text.
